# Quercetin-Rich Ethanolic Extract of *Polygonum odoratum* var Pakphai Leaves Decreased Gene Expression and Secretion of Pro-Inflammatory Mediators in Lipopolysaccharide-Induced Murine RAW264.7 Macrophages

**DOI:** 10.3390/molecules27123657

**Published:** 2022-06-07

**Authors:** Nittaya Chansiw, Sorraya Champakam, Pattranuch Chusri, Kanjana Pangjit, Somdet Srichairatanakool

**Affiliations:** 1School of Medicine, Mae Fah Luang University, Chiang Rai 57100, Thailand; nittaya.cha@mfu.ac.th (N.C.); pattranuch.chu@mfu.ac.th (P.C.); 2School of Integrative Medicine, Mae Fah Luang University, Chiang Rai 57100, Thailand; sorraya@mfu.ac.th; 3College of Medicine and Public Health, Ubon Ratchathani University, Ubon Ratchathani 34190, Thailand; kanjana.pa@ubu.ac.th; 4Oxidative Stress Cluster, Department of Biochemistry, Faculty of Medicine, Chiang Mai University, Chiang Mai 50200, Thailand

**Keywords:** anti-inflammation, phenolics, *Polygonum odoratum*, pro-inflammatory mediators, quercetin, RAW264.7 cells

## Abstract

*Polygonum odoratum* var. Pakphai has been used in traditional Thai medicine for the treatment of flatulence and constipation and to relieve the inflammation caused by insect bites. Quercetin (Q), which is abundant in plant-based foods, has been found to exert anti-inflammatory properties. This study evaluated the anti-inflammatory activity of *P. odoratum* ethanolic extract in RAW264.7 macrophage cells. Leaves were extracted with 50% ethanol, phenolics and flavonoids were then analyzed using UHPLC-QTOF-MS and HPLC-DAD. RAW264.7 cells were induced with lipopolysaccharides (LPSs). They were then treated with the extract and prostaglandin E_2_ (PGE_2_), and interleukin-6 (IL-6) and tumor necrotic factor-alpha (TNF-α) concentrations were determined. Levels of cyclooxygenase-2 (COX-2), inducible nitric oxide synthase (iNOS), IL-6 and TNF-α mRNAs were analyzed using qRT-PCR. Chemical analysis demonstrated that the extract was abundant with Q while also containing catechin, gallic acid, epicatechin gallate and coumarin. The extract increased the viability of RAW264.7 cells and dose-dependently decreased nitric oxide production, PGE_2_, IL-6 and TNF-α levels in the medium from the LPS-induced RAW264.7 cell culture. Consistently, COX-2, iNOS, IL-6 and TNF-α mRNA levels were decreased in a concentration-dependent manner (*p* < 0.05). Thus, the quercetin-rich ethanolic extract derived from *P. odoratum* var Pakphai leaves can exert anti-inflammatory activity in LPS-induced RAW264.7 cells through a reduction of the pro-inflammatory mediator response.

## 1. Introduction

The current state of inflammation is recognized as an immune process that responds to specific stimuli, such as certain microorganisms, tissue injuries and the presence of certain chemicals [1]. Inflammatory processes involve vascular changes and white blood cell responses. Macrophages are the main phagocytotic cells that respond by releasing several pro-inflammatory mediators, such as tumor necrosis factor-alpha (TNF-α), interleukin-1beta (IL-1β), interleukin-6 (IL-6) and other chemo-attractants, to eliminate noxious stimuli and assist in repairing the affected tissue [2]. In addition, TNF-α activates the expression and synthesis of inducible nitric oxide synthase (iNOS), resulting in an increased production of nitric oxide radicals (NO^•^). During the inflammation process, cyclooxygenase-2 (COX-2) catalytically converts arachidonic acid to prostaglandin E_2_ (PGE_2_), a main inflammatory mediator that is associated with pain, swelling, redness and heat [3]. Importantly, inflammation is a major factor for the progression of various chronic diseases, including diabetes, cancer, cardiovascular disease, arthritis, obesity, autoimmune diseases and inflammatory bowel disease [4,5].

*Polygonum odoratum* Lour. (Family Polygonaceae), which has been reclassified as *Persicaria odorata*, was originally cultivated by the Hmong people in China and Laos for cooking with fish and exotic herbs [6]. It is a biennial edible plant that has a hot, spicy taste and a strong coriander-like aroma. In addition, it is usually used to add flavor and aroma to various Thai dishes. Other synonyms currently being used to refer to this plant include rau ram or sang hum in Viet Nam, where it is referred to as mint or coriander in Vietnamese cuisine. It is also known as Daun Laksa or the Laksa plant in Malaysia. Furthermore, it is known as Paew in the northeast and Pakphai in the northern regions of Thailand. Interestingly, the plant leaves are commonly used as an antidiabetic, anti-microbial, anti-inflammatory and anti-tumor agent in Asian traditional medicine. It has been used in Thai traditional medicine for the treatment of flatulence, constipation and the localized inflammation caused by insect bites [7]. Nowadays, qualitative analysis involving high-performance liquid chromatography coupled with electrospray ionization-mass spectrometry (HPLC–ESI-MS), comprehensive analysis, such as ultrahigh-performance liquid chromatography coupled with electrospray ionization-quadrupole time-of-flight-mass spectrometry (UHPLC-ESI-QTOF-MS/MS), and quantitative analysis, such as high-performance liquid chromatography-diode array detection (HPLC-DAD) are rapid and sensitive liquid chromatographic techniques that can be used to identify major chemical components (such as phenolics and flavonoids) and secondary metabolites in natural and extracted plant parts. Accordingly, the obtained samples are tested for toxicity or investigated for their biological and pharmacological activities [8,9,10,11,12,13,14].

Evidence suggests that the ethanolic extracts of the aerial sections of *P. odoratum* contain scutellarein-7-glucoside and quercitrin, while exhibiting anti-inflammatory activity in lipopolysaccharide (LPS)-induced murine RAW264.7 macrophage cells [15]. Moreover, a previous study has recently reported that *P. odoratum* leaf extract abundant with phenolic compounds can exert strong antioxidant, anti-cancer and anti-microbial activities [16]. In product formulations, alginate-encapsulated *P**. odoratum* ethanolic extract beads were found to possess an efficient anti-bacterial property by completely inhibiting *Staphylococcus aureus* and *Escherichia coli* growth in agar [17]. However, the anti-inflammatory effects associated with the pro-inflammatory response are still unclear. The present study focused on analyzing the major active phenolic compounds in the ethanolic extracts of *P. odoratum* var. Pakphai leaves, evaluating the toxicity in murine RAW264.7 macrophages and investigating the anti-inflammatory activity in LPS-induced RAW264.7 cells.

## 2. Results

### 2.1. Extraction Yield, Total Phenolic and Flavonoid Contents

Interestingly, in terms of the mean ± standard error of the mean (SEM) values, 50% of ethanolic extractions of *P. odoratum* cv. Pakphai leaves produced an average yield of 18.96% (*w*/*w*), for which the total phenolic content (TPC) and total flavonoid content (TFC) values were 52.59 ± 0.58 mg gallic acid equivalent (GAE)/g extract and 19.97 ± 0.11 mg quercetin equivalent (QE)/g extract, respectively.

### 2.2. UHPLC-ESI-QTOF-MS/MS Profiling of the Extract

MS experiments on the UHPLC-MS system coupled with the ESI source, QTOF and an ion trap mass spectrometer were carried out in order to investigate and identify the presence of different isobaric compounds and to perform a comprehensively qualitative analysis of the phenolic constituents present in 50% ethanolic extract of *P. odoratum* Lour. cv. Pakphai leaves. The ESI–MS base peak chromatogram of the extract shows a relatively complex mixture containing peak quantities of phenolics, flavonoids and lignans. Apparently, these two phenolics included epicatechin (EC) at T_R_ 17.312 min and gallocatechin (GC) at T_R_ 18.242 min. Two flavonoids included quercetin 3′-*O*-glucuronide (QG) at the time of retention (T_R_) at 9.455 min and 5,7,8,3′,4′-pentahydroxyisoflavone or quercetin (Q) at T_R_ 14.787 min, and one lignan included 3-methylellagic acid 8-rhamnoside at T_R_ 11.016 min. These were identified using fragmentation patterns observed in tandem mass spectra (Figure 1). Accordingly, estimated and exact mass formulae and the chemical structures of the possible compounds are presented in Table 1. Interestingly, hesperidin (T_R_ 21.531 min), which is a flavanone glycoside found in citrus fruits whose aglycone form is called hesperetin, was detected in the extract. However, six other peaks eluted at T_R_ values of 0.356, 0.677, 0.913, 2.112, 17.112 and 22.146 min could not be identified herein due to a lack of pertinent information and an established database.

### 2.3. HPLC-DAD Analysis of Phenolic Compounds in the Extract

For the purposes of calibration, different concentrations of mixed standard phenolics (10 μL) were injected into the HPLC system to indicate the relative phenolic compounds present in the extract eluted from the column and to establish the standard curves needed to calculate the appropriate concentrations. In our findings, the standard GA, C, EGCG, EC, ECG, CO and Q quantities came out at specific T_R_ values of 7.02, 12.27, 14.00, 15.40, 17.28, 21.27 and 26.48 min, respectively. Correspondingly, GA, catechin (C), epicatechin 3-gallate (ECG), coumarin (CO) and Q were detected in the ethanolic extract of *P. odoratum* Lour. cv. Pakphai leaves (Figure 2A), while epigallocatechin 3-gallate (EGCG) and EC were not detected (Figure 2B). Accordingly, concentrations of the five phenolics were determined from the standard curves and have been expressed in Table 2, of which Q was the most abundant and ECG was the second most abundant.

### 2.4. Toxic Effect on RAW264.7 Macrophage Cells

Clearly, all concentrations of the extracts exerted no cytotoxic effects on RAW264.7 macrophages. Concentrations of 25–200 μg/mL significantly increased cell viability in a concentration-dependent manner, while concentrations of >200 μg/mL did not have any influence (Figure 3).

### 2.5. Effect on Nitric Oxide Production and Inducible Nitric Oxide Synthase Gene Expression

In the present study, LPS treatment markedly increased levels of NO^•^ production in RAW264.7 macrophages when compared with control cells without treatment (*p* < 0.05); nevertheless, increasing NO^•^ values were dose-dependently reduced in macrophages following treatment with the extract (100, 200 and 400 μg/mL) with a half maximal inhibitory concentration (IC_50_) value of >400 μg/mL) (*p* < 0.05) (Figure 4A). Moreover, iNOS mRNA levels were evidently increased in LPS-induced cells (*p* < 0.05) when compared with control cells without LPS induction. Consistently, the extracts were found to decrease in accordance with the increasing iNOS mRNA expression in a concentration-dependent manner (*p* < 0.05) when compared with the cells that did not undergo extract treatment (Figure 4B).

### 2.6. Inhibitory Effects on Cyclooxygenase 2 Gene Expression and PGE_2_ Production

In terms of the relevant pharmacological effects, cyclooxygenases (COXs) comprise constitutive COX or cyclooxygenase 1 (COX-1) and inducible COX (COX-2), which catalyze a conversion of arachidonic acid to prostaglandins, in particular PGE_2_, resulting in inflammation. Herein, we have investigated whether the ethanolic extract would exert inhibitory effects on COX-2 gene expression and PGE_2_ production in LPS-induced murine RAW264.7 macrophages, indicating an anti-inflammatory activity. Thus, we have assessed levels of COX-2 mRNA expression in the macrophages with the use of qRT-PCR and PGE2 production in the culture medium using ELISA. According to our findings, COX-2 mRNA expression in LPS-induced macrophages was highly upregulated by LPS stimulation (*p* < 0.05), while increases in COX-2 mRNA were dose-dependently decreased by treatment with the extract (*p* < 0.05) (Figure 5). Consistently, the results provide evidence that PGE_2_ levels were significantly increased in LPS-induced RAW264.7 macrophages when compared with control cells without LPS induction. The treatment with the extract decreased PGE_2_ production in a concentration-dependent manner with an IC_50_ value of 231.9 μg/mL) (*p* < 0.05) when compared with LPS-induced cells without the extract treatment (Figure 6).

### 2.7. Effects on Production and Gene Expression of Pro-Inflammatory Cytokines

LPS stimulation was found to increase levels of IL-6 concentrations and mRNA (*p* < 0.05) in murine RAW264.7 macrophages when compared with the control cells; however, treatment with the extract significantly lowered the increasing IL-2 levels in a concentration-dependent manner (Figure 7A,B). Similarly, LPS also increased levels of TNF-α concentrations and mRNA (*p* < 0.05) when compared with the control cells, while treatment with the extract significantly decreased the production and gene expression of TNF-α in a concentration-dependent manner with an IC_50_ value of 285.3 μg/mL) (Figure 7C,D). The results indicate the anti-inflammatory effects of ethanolic extract of *P. odoratum* Lour. cv. Pakphai leaves in macrophages.

## 3. Discussion

Traditionally, the *P. odoratum* Lour. plant was cultivated in urban gardens by members of Hmong hilltribes in Lao who typically produced edible vegetables with high mineral profiles (e.g., Ca, Mg and Mn), various fish seasonings, natural components of suggested pregnancy and post-partum diets and formulated a variety of medicinal herb recipes [6]. Giving it its aromatic odor, the aerial parts of this plant contain many volatile organic compounds, including (Z)-3-hexenal, (Z)-3-hexenol, decanal, undecanal, dodecanal, 3-sulfanyl-hexanal and 3-sulfanyl-hexan-1-ol [18]. Due to its pungent taste, the leaves of this plant contain various active compounds, such as polygodial. Furthermore, the leaves have been determined to contain 1,4-dialdehyde, which is derived from drimane terpenoids [18]. Additionally, fresh *P. odoratum* leaves were found to contain a range of essential oils that comprise twenty-five compounds, of which dodecanal, decanal and anisaldehyde were the most abundant and exhibited an inhibitory effect on tyrosine activity and *Salmonella choleraesuis* growth [16,19]. Our previous study has demonstrated that only the methanol extract of *P. odoratum* var. Pakphai leaves were abundantly present in phenolics and flavonoids together with E-15-heptadecenal and 3,7,11,15-tetramethyl-2-hexadecen-1-ol, which exhibited strong free-radical scavenging properties [7]. The chromatogram of the leaf extract shows the chemical composition of QG, 5,7,8,3′,4′-pentahydroxyisoflavone and 3-methylellagic acid 8-rhamnoside, of which QG was found to possess potent anti-cancer [20], anti-aging [21] and antioxidant properties [22]. A number of previous studies have reported on the anti-inflammatory properties of QG that suppressed a pro-inflammatory mediator response on LPS-stimulated RAW264.7 macrophage cells [23,24] and inhibited vascular permeability in mice [25]. Likewise, 5,7,8,3′,4′-pentahydroxyisoflavone or quercetin is an isoflavone derivative that has been reported to possess remarkable anti-inflammatory effects [26] and anti-proliferative activities [27]. Our present study also produced a polyphenolic profile using HPLC-DAD and UHPLC-ESI-QTOF-MS/MS methods. It has been confirmed that 50% (*v*/*v*) ethanolic extract of the plant leaves contain high levels of quercetin and have also been found to contain epicatechin gallate, coumarin, gallic acid and catechin.

With regard to the potential health benefits of this plant, the leaves contain high amounts of bioactive ingredients that exhibit antioxidant, anti-hemolytic and anti-bacterial properties. As an alternative form of osteoporosis therapy, the oral administration of *Morus alba* and *P. odoratum* leaves over a period of 3 months effectively decreased levels of bone oxidative stress markers and osteoclast density but elevated serum calcium, alkaline phosphatase (ALP) and osteocalcin levels. The administration also increased osteoblast density and cortical thickness in treated ovariectomized rats [28]. Consistently, the data obtained from a randomized double-blind placebo-controlled clinical trial has supported the contention that the consumption of *M*. *alba* leaf extract (50 mg/day) combined with *P*. *odoratum* leaf extract (1500 mg/day) for 8 weeks could elevate serum levels of ALP, osteocalcin and TPC but could effectively reduce serum beta-isomerized C-terminal telopeptide levels in menopausal subjects [29]. We have previously demonstrated that water, dichloromethane and the methanolic extracts derived from Pakphai leaves were not toxic to RAW264.7 macrophages and exhibited an anti-inflammatory activity in LPS-induced RAW264.7 cells, of which an IC_50_ value of dichloromethane = 53.75 ± 0.75 μg/mL was achieved by inhibiting the production of NO^•^ [7]. The present study using a colorimetric MTT test has determined that extract doses at 100–400 μg/mL were not toxic to RAW264.7 cell cultures but seemed to increase cell viability. Similarly, Okonogi and coworkers reported on the nontoxic effect of the ethanolic extract fractions (10–100 μg/mL) that included scutellarein-7-glucoside and quercitrin on RAW264.7 cells. Importantly, the doses they administered were in much lesser quantities than the ones we had used in our experiments [15]. Although macrophages play essential roles in anti-inflammatory defense mechanisms, the abnormal activation of macrophages has been reported in the development of many inflammatory disorders, including sepsis, rheumatoid arthritis, inflammatory bowel disease and cancer. Under pathogenic conditions, an excessive amount of pro-inflammatory mediators and cytokines are produced from abnormally activated macrophages that eventually provoke an inflammatory response [30]. Therefore, the inhibition of abnormal macrophage activation might be an invaluable therapeutic goal in the treatment of inflammatory disorders.

Several previous studies have reported on the anti-inflammatory activity facilitated by quercetin, which was found to be abundant in the ethanolic extracts of various plant-derived products and compounds. The ethanolic extract of *Myrsine seguinii* is rich in Q, which was found to inhibit inflammatory responses (such as production of NO^•^ and PGE_2_) in LPS-stimulated RAW264.7 cells and LPS-induced mouse peritonitis by blocking the Src/Syk/nuclear factor kappa B (NF-κB) and the IL-1 receptor-associated kinase 1/activator protein 1 (IRAK-1/AP-1) pathways [31]. Additionally, 50% ethanolic extract of persimmon leaves potently inhibited the production of NO^•^, PGE_2_ and IL-6 in LPS-induced RAW264.7 cells [32]. Likewise, the ethanolic extract of the *Euphorbia kansui* root containing ingenane diterpenoids (euphorkans A and B), together with 16 known analogues, exerted an anti-inflammatory activity that was consistent with Q by inhibiting the effects of certain inflammation mediators, such as TNF-α and IL-6, in a concentration-dependent manner through the inhibition of NF-κB activity [33]. Moreover, the ethanolic extract of the QingXiaoWuWei decoction, containing quercetin and other ingredients, exhibited synergistic anti-inflammatory activity in the LPS-induced RAW264.7 cell culture via JUN, MAPK1 and AKT1 targets and the mitogen-activated protein kinases (MAPK)/phosphatidylinositol-3 kinase (PI3K)/Akt serine/threonine kinases pathways [34].

In this study, we found that the ethanolic leaf extracts of this plant contain high amounts of phenolic compounds and flavonoids. Notably, our ethanolic extract of *P. odoratum* leaves (100–400 μg/mL) significantly attenuated pro-inflammatory mediators and cytokine production, such as NO^•^, PGE_2_, IL-6 and TNF-α, in LPS-induced RAW264.7 cells in a concentration-dependent manner. Consistently, the extract also down-regulated the gene expression of iNOS, COX-2, IL-6 and TNF-α. In contrast, secretion of IL-6, but not TNF-α, was found to be significantly reduced in a concentration-dependent manner with 50% and 100% ethanolic extracts (10–100 μg/mL) of *P. odoratum* leaves (IC_50_ = 25 μg/mL), the scutellarein-7-glucoside fraction (IC_50_ = 102 μM) and the quercitrin (IC_50_ = 77 μM) extract, all of which emphasized the anti-inflammatory activity of scutellarein-7-glucoside and quercitrin [15]. Moreover, the hot water extract of *P. odoratum* leaves, which are known to be abundant with TPC (223.0 ± 9.7 mg GAE/mg extract) and comprise GA (7.63 ± 0.30 mg/g), ellagic acid (3.31 ± 0.14 mg/g), C (1.77 ± 0.10 mg/g), rutin (0.40 ± 0.20 mg/g), Q (0.08 ± 0.01 mg/g), caffeic acid (0.07 ± 0.01 mg/g), kaemferol (0.02 ± 0.00 mg/g) and ferulic acid (0.02 ± 0.00 mg/g), exerted antioxidant activities by decreasing malondialdehyde levels while increasing levels of catalase, superoxide dismutase and glutathione peroxidase activities in RAW264.7 cells exposed to X-ray irradiation [35]. Furthermore, the water extract of *P. odoratum* leaves decreased contractions of 80 mM potassium chloride-induced rat ileum in a dose-dependent manner, possibly by increasing the production of NO^•^ via the β-adrenergic receptor pathway and by blocking the calcium influx [36].

Herein, our findings suggest the potent anti-inflammatory effect of specific major active compounds, particularly quercetin and isoflavone derivatives, in the ethanolic extracts of *P. odoratum* var. Pakphai leaves through suppression in the responses of certain pro-inflammatory mediators, such as NO^•^, PGE_2_, IL-6 and TNF-α in LPS-stimulated murine macrophage cells. The underlying mechanism of quercetin on the pro-inflammatory mediator in macrophages has been demonstrated, wherein LPS activates MAPKs, such as extracellular signal-regulated kinases, c-Jun N-terminal kinase and p38 MAPK [37]. In addition, the transcription factors that occur under LPS-stimulated inflammatory conditions are downstream targets of the MAPK pathways, which are known to regulate various gene encoding inflammatory mediators. The ethanolic extract reinforces the anti-inflammatory activity exerted by some of the other extracts that had been previously used. This confirmed the presence of a more lipophilic phenolic compound. Accordingly, quercetin was found to participate in the potent anti-inflammatory response in the macrophage cell culture in this study and in animals included in other related studies.

## 4. Materials and Methods

### 4.1. Chemicals and Reagents

Dulbecco’s modified Eagle medium (DMEM) and fetal bovine serum (FBS) were purchased from Gibco, Thermo Fisher Scientific, Waltham, MA, USA. Penicillin–Streptomycin, lipopolysaccharide (derived from *Escherichia coli*, 3-(4,5-dimethyl-2-thiazolyl)-2,5-diphenyl-2-H-tetrazolium bromide (MTT), N-(1-naphthyl) ethylene dihydrochloride, sulfanilamide, 6-hydroxy-2,5,7,8-tetramethylchroman-2-carboxylic acid (Trolox), phosphate buffered saline pH 7.0 (PBS), dimethylsulfoxide (DMSO), 0.25% trypsin-EDTA solution containing 2.5 g porcine trypsin and 0.2 g EDTA sodium salt/L of Hanks’ balanced salt solution with phenol red) and Griess reagent (containing 0.1% (*w*/*v*) naphthyl-ethylene-diamide dihydrochloride, 1% (*w*/*v*) sulfanilamide and 5% (*v*/*v*) phosphoric acid) were purchased from Sigma-Aldrich Chemicals Company, St. Louis, MO, USA. Standard GA, C, EGCG, EC, ECG, CO and Q were purchased from the Sigma-Aldrich Chemicals Company, St. Louis, MO, USA. All organic solvents (HPLC or HPLC/MS grade) were of the highest grade. 

### 4.2. Cell Culture

The murine macrophage (RAW264.7) cell line was purchased from the American Type Culture Collection (ATCC, TIB-71, VA, USA) and was cultured in DMEM, supplemented with 10% (*v*/*v*) FBS with 100 U/mL penicillin and 100 µg/mL streptomycin (DMEM^+^) and then maintained at 37 °C in a 5% CO_2_ incubator. The cell confluent was expected to be 70–80% after harvesting.

### 4.3. Preparation of P. odoratum Ethanolic Extract

*Polygonum odoratum* leaves were freshly harvested from a *P. odoratum* field located at Mae Fah Luang University, Chiang Rai, Thailand, and identified by Dr. Jantrararuk Tovaranon, PhD. at the School of Science, Mae Fah Luang University, Chiang Rai. The herbarium specimens (Herbarium number: MD2018080001-1) were prepared and deposited in the Medicinal Plant Innovation Center of Mae Fah Luang University. The leaves were dried using a shade drying method and ground into a fine powder with a mechanic milling machine (Wells-Index, Muskegon, MI, USA). The dried powder (50 g) was extracted with 50% (*v*/*v*) ethanol (750 mL) at room temperature for 72 h using an orbital shaker. The supernatant was filtered through Whatman’s No.1 filter paper (polyethersulfone-type, Whatman International Limited Company, Maidstone, UK). The ethanolic filtrate was evaporated to the point of dryness using a freeze-drying machine (Labconco™, Labcono Corporation, MO, USA). The extract was divided in aliquots in dark vials and kept in a freezer at −20 °C. They were then reconstituted in 0.1% DMSO solution before being used.

### 4.4. Determination of TPC

The TPC of the extract was determined using the Folin–Ciocalteu method with slight modifications [38]. Briefly, the extract solution (0.1 mL) was mixed with working Folin–Ciocalteu reagent (1:1 dilution) (1.0 mL) and 7% (*w*/*v*) sodium bicarbonate (1.0 mL). It was incubated at room temperature for 30 min and an absorbance (A) value of 765 nm was measured against the reagent using a Shimadzu UV-1800 UV/VIS spectrophotometer (Cole-Parmer, Vernon Hills, IL, USA). The TPC was calculated from the standard curve constructed using standard GA at concentrations of 10–100 µg/mL and expressed as mg GAE/g extract.

### 4.5. Determination of TFC

TFC was determined using an aluminum chloride colorimetric method [39]. Briefly, the extract solution (0.1 mL) was mixed with 10% (*v*/*v*) aluminum chloride solution (1.0 mL) and 1 M potassium acetate solution (1.0 mL). It was then incubated at room temperature for 40 min, and an A value of 415 nm was measured against a blank reagent using a Shimadzu UV-1800 UV/VIS spectrophotometer (Cole-Parmer, Vernon Hills, IL, USA). The TFC was calculated from the standard curve constructed using standard Q at concentrations of 10–200 µg/mL and expressed as mg of the QE/g extract.

### 4.6. UHPLC-ESI-QTOF-MS/MS Analysis

The extract (1 mg) was initially dissolved in methanol (1 mL), then 10-fold diluted to reach a concentration of 1 µg/mL and manually filtered through a syringe membrane filter (Whatman’s polytetrafluoroethylene type, 0.22-μm pore size, Sigma-Aldrich Chemicals Company, St. Louis, MO, USA). The filtrate was transferred to HPLC autosampler vials and the mode of the phytochemicals was analyzed using the UHPLC–ESI-QTOF-MS/MS technique [27]. The system included the use of an UHPLC machine (Agilent 6500 Series, Agilent Technologies, Santa Clara, CA, USA) equipped with an ESI source employing orthogonal nebulization and a heated counterflow drying gas system to achieve excellent sensitivity and robust, reliable degrees of performance. The chromatographic instrument was connected in series to a DAD (Agilent 1260 Infinity II Series, Agilent Technologies, Santa Clara, CA, USA) and an MS detector (Agilent 6550 Series, Agilent Technologies, Santa Clara, CA, USA). The MS detector was set up for full scanning of the mass spectra from *m/z* 100 to 1000 g/mole and employed in positive ion mode with a nebulizer. The gas temperature was set at 350 °C and gas flow was set to 13 L/min. In the analysis process, the sample (1.0 µL) was automatically injected into the machine and fractionated on the column (Zorbax eclipse plus C18 type, dimension 2.1 mm × 50 mm, 1.7 µm particle size, Agilent Technologies, Santa Clara, CA, USA) and eluted by the mobile phase comprising solvent A (deionized water containing 0.1% formic acid) and solvent B (acetonitrile containing 0.1% formic acid). The gradient elution was performed at a flow rate of 400 µL/min for a total running time of 26 min, for which the program was set up to start at 5% solvent B for 1 min, then increased to 17% solvent B within 13 min and increased up to 100% solvent B within 20 min. The 100% solvent B elution was maintained in order to wash the column for 2 min before it was decreased to 5% solvent B over a period of 3 min. Analysis of possible compounds was performed using Agilent Mass hunter B 8.0 software (Qualitative navigator, Qualitative workflows) and the PCDL database. Peak identification was performed by comparing the retention time, mass spectra and fragmentation patterns with reference to the compounds obtained from various data libraries and networks (such as ChemSpider).

### 4.7. HPLC-DAD Analysis

The extract (1 mg) and standard phenolics, including GA (0.625–20 μg/mL), C (3.12–100 μg/mL), EGCG (1.56–50 μg/mL), EC (3.12–100 μg/mL), ECG (0.78–25 μg/mL), CM (0.625–20 μg/mL) and Q (0.625–20 μg/mL), were dissolved in 50% (*v*/*v*) ethanol (1 mL) and filtered through a membrane filter (Whatman’s polytetrafluoroethylene type, 0.45-μm pore size, Sigma-Aldrich Chemicals Company, St. Louis, MO, USA) and analyzed using the HPLC-DAD technique [40]. The HPLC system (Thermo Scientific Dionex UltiMate 3000 Series, Thermo Fisher Scientific Inc., Waltham, MA, USA) comprised quaternary pumps, an autosampler, a thermo-stated column oven and DAD. For the quantitation of phenolic compounds, the extract (10 μL) was automatically injected into the HPLC system, fractionated on the column (Dionex IonPac AmGC18-type, dimension 4.6 mm × 250 mm, 5 µm particle size, Thermo Fisher Scientific Inc., Waltham, MA, USA), thermally regulated at 25 °C and eluted with two mobile-phase solvents, namely, A (0.1% trifluoroacetic acid in water) and B (methanol). The linear gradient elution was prepared as follows: 90% A: 10% B (0 → 35 min), 10% A: 90% B (35 → 40 min) and 90% A: 10% B (40 → 45 min) at a flow rate of 1.0 mL/min and dual detection wavelengths of 254 and 280 nm. Eluents were identified by comparing their retention time (T_R_) values with those of the authentic standards. Concentrations of the relative phenolic compounds were determined using the curves of the authentic standards, including GA, C, EC, ECG and EGCG. They were then expressed as μg/mg of the dry extract.

### 4.8. Cell Viability Assay

Cell viability was determined using the MTT method [41]. Briefly, RAW264.7 cells (5 × 10^3^/well) were cultured in DMEM^+^ and seeded in a 96-well plate until reaching 80% confluence. The cells were treated with the extract (0–100 µg/mL) for 24 h, washed with PBS and incubated with the MTT solution (5 mg/mL in PBS) at 37 °C for another 4 h. Finally, 0.1% (*w*/*v*) DMSO solution was added to dissolve the formazan product and the A value was measured at a wavelength of 540/630 nm against the blank reagent using a Shimadzu UV-1800 UV/VIS spectrophotometer (Cole-Parmer, Vernon Hills, IL, USA). The percentage of viable cells after the treatment was calculated when compared with the untreated cells representing 100% viable cells.

### 4.9. Assessment of Pro-Inflammatory Mediators

RAW264.7 cells were cultured in DMEM^+^ at 37 °C for 24 h and seeded in 24-well plates at a density of 5 × 10^4^ cells/well. Afterwards, the cells were cultured in DMEM^+^ with or without LPS (2 µg/mL) at 37 °C for 24 h, treated with the extract (100, 200 and 400 μg/mL) at 37 °C for another 24 h and harvested using trypsin-EDTA solution. After centrifugation at 12,000× *g* rpms for 10 min, the supernatant was assayed using NO^•^, PGE_2_, IL-6 and TNF-α concentrations as has been described below.

### 4.10. Determination of Nitric Oxide Concentrations

The supernatant was incubated with Griess reagent at room temperature for 15 min and the A value was measured at 650 nm against a blank reagent using a Shimadzu UV-1800 UV/VIS spectrophotometer (Cole-Parmer, Vernon Hills, IL, USA) [42]. NO^•^ concentrations were calculated from the standard curve established from different concentrations of sodium nitrite.

### 4.11. Determination of PGE_2_, IL-6 and TNFα Concentrations

PGE_2_ was quantified using competitive ELISA (Product number ADI-900-001), IL-6 using sandwich ELISA (Product number ADI-900-045) and TNFα using sandwich ELISA (Catalogue number E-EL-M3063) kits according to the manufacturer’s instructions (ENZO, Elabscience, Houston, TX, USA) [43,44].

### 4.12. Quantitative Real-Time Polymerase Chain Reaction

Total RNA was extracted using the PureLink™ RNA Mini Kit (Invitrogen Life Sciences, Carlsbad, CA, USA) according to the manufacturer’s instructions. Complementary DNA (cDNA) was synthesized from 1 μg of total RNA using a SensiFASTTM cDNA synthesis kit (Bioline Reagent, London, UK). Accordingly, qRT-PCR was performed using a SensiFASTTM SYBR No-ROX kit (Bioline Reagent, London, UK) and a reaction mixture that consisted of SYBR Green 2 × PCR Master Mix and a cDNA template, as well as the use of forward and reverse primers. The PCR protocol included an initial hold at 95 °C for 2 min, followed by a 2-step PCR program of 95 °C for 15 s and 58 °C for 30 s for 39 cycles [23]. The relative levels of iNOS, COX-2, IL-6 and TNFα mRNA expressions were normalized to that of glyceraldehyde 3-phosphate dehydrogenase (GAPDH) using the 2ΔΔCT method. The primer sequences used in this study are shown in Table 3.

### 4.13. Statistical Analysis

Data were analyzed using IBM SPSS Statistics Program version 2.2 and are expressed as the mean ± SEM values of three independent experiments. Statistical significance was determined using a one-way analysis of variance (ANOVA) post hoc Tukey’s test, for which a *p* value < 0.05 was considered significantly different.

## 5. Conclusions

This study highlighted the determination that the ethanolic extract of *Polygonum odoratum* Lour. cv. Pakphai leaves contain, bioactive phenolics such as gallic acid, catechin and epicatechin 3-gallate, as well as flavonoids with a high content of quercetin and coumarin. These substances are known to increase the viability of RAW264.7 macrophage cells. Importantly, the extract inhibited iNOS gene expression while decreasing nitric oxide production. Furthermore, an increase in COX-2 activity was observed, along with a decrease in PGE2 production and reduced IL6 and TNFα production and gene expression in LPS-induced RAW264.7 cells. The findings indicate the anti-inflammatory activity of this plant through the attenuation of a pro-inflammatory mediator response in macrophage cells. In terms of its potential value-added properties, the ethanolic extract appears to be a nutraceutical product that could be used as an alternative form of treatment or a complementary treatment for inflammatory diseases. In further studies, we will intensively investigate the anti-inflammatory role of Pakphai ethanolic extract in allergic asthma and its underlying mechanisms in rats challenged by the environmental allergen ovalbumin and LPS. Additionally, gastric antiulcer activity of the extract will be assessed in ethanol/hydrochloric acid-induced mice and rats.

## Figures and Tables

**Figure 1 molecules-27-03657-f001:**
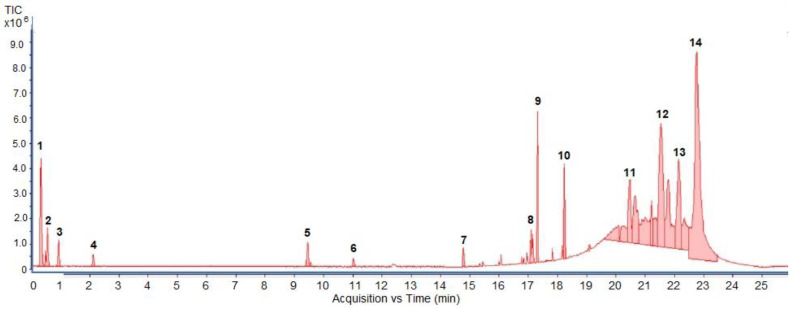
UHPLC–ESI-QTOF-MS/MS extracted ion chromatogram produced from 50% ethanolic extract of *P. odoratum* Lour. cv. Pakphai leaves at the selected deprotonated molecules [M—H]^−^. Abbreviations: TIC = total ion counts, UHPLC-ESI-QTOF-MS/MS = ultrahigh-performance liquid chromatography–electrospray ionization-quadrupole time-of-flight-mass spectrometry/mass spectrometry.

**Figure 2 molecules-27-03657-f002:**
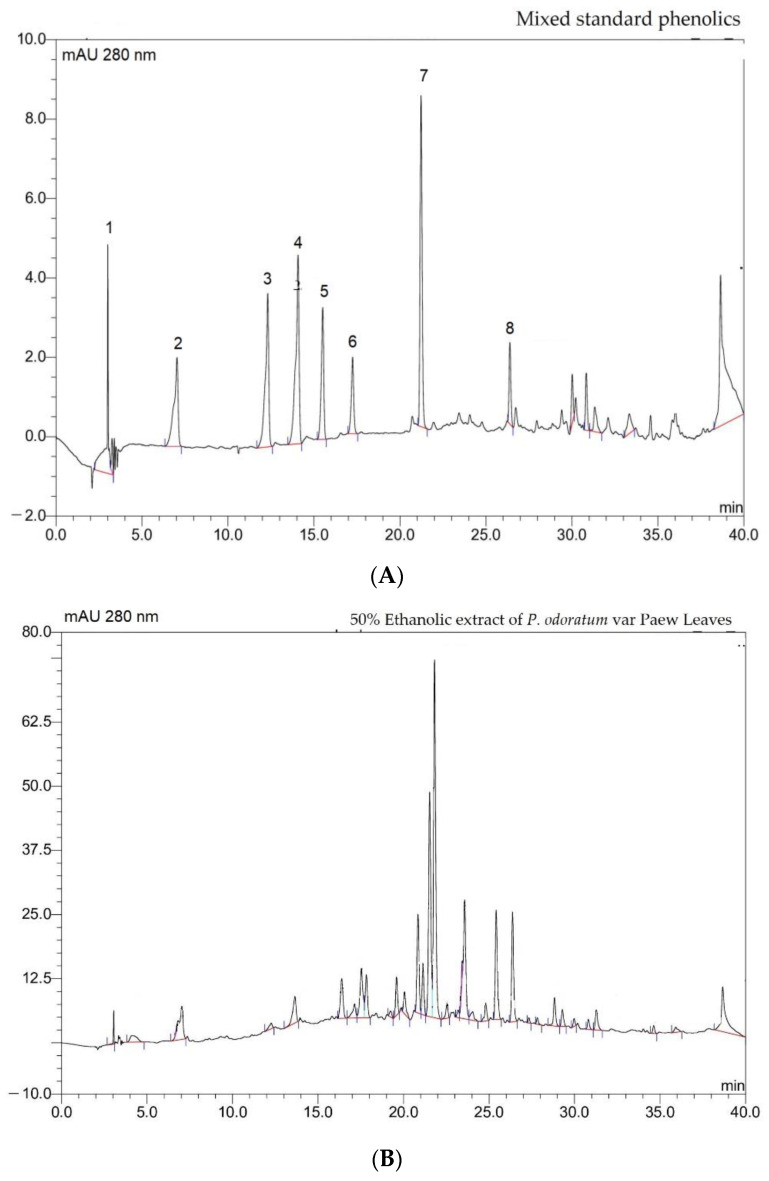
HPLC-DAD profile of phenolic compounds in the mixed standards that included gallic acid, catechins, epigallocatechin 3-gallate, epicatechins, epicatechin gallate, 20 μg coumarin, quercetin (**A**) and 50% ethanolic extract of *P. odoratum* Lour. cv. Pakphai leaves (**B**).

**Figure 3 molecules-27-03657-f003:**
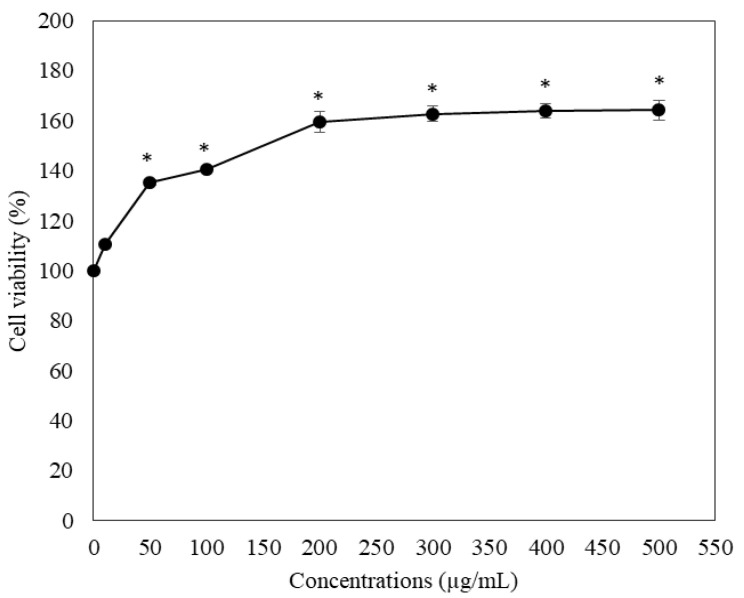
MTT test-based viability of RAW264.7 macrophage cells treated with 50% ethanolic extracts of *P. odoratum* Lour. cv. Pakphai leaves. Data obtained from three independent experiments are expressed as mean ± SEM values. Accordingly, * *p* < 0.05 was considered significant when compared with untreated cells. MTT = 3-(4,5-dimethyl-2-thiazolyl)-2,5-diphenyl-2-H-tetrazolium bromide, SEM = standard error of the mean values.

**Figure 4 molecules-27-03657-f004:**
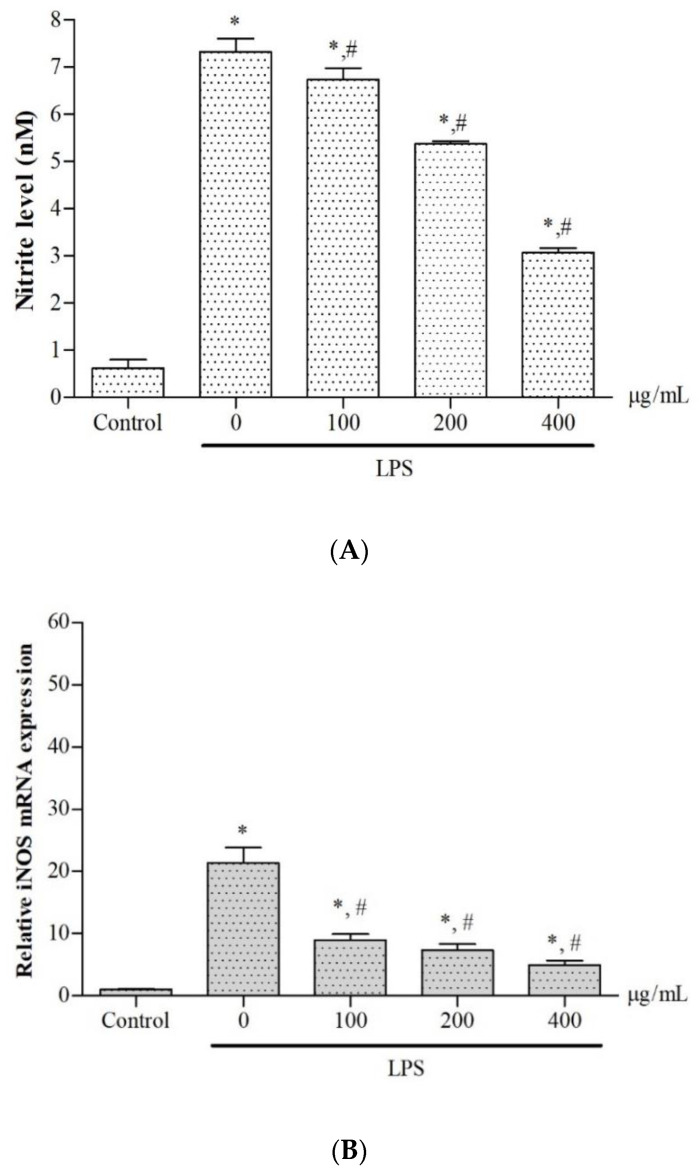
Levels of NO^•^ production (**A**) and iNOS gene expression (**B**) in LPS-induced RAW264.7 macrophages after treatment with 50% ethanolic extract of *P. odoratum* Lour. cv. Pakphai leaves (100, 200 and 400 μg/mL) for 24 h. Data obtained from three independent experiments are expressed as mean ± SEM values. Accordingly, * *p* < 0.05 was considered significant when compared to cells without LPS induction; # *p* < 0.05 was considered significant when compared to LPS-induced cells without the extract treatment. Abbreviations: iNOS = inducible nitric oxide synthase, LPS = lipopolysaccharide, NO^•^ = nitric oxide radical, SEM = standard error of the mean values.

**Figure 5 molecules-27-03657-f005:**
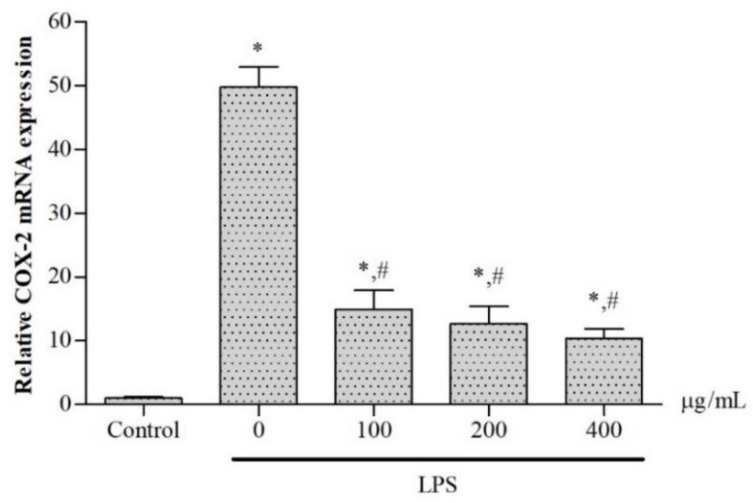
Levels of COX-2 mRNA expression in LPS-induced RAW264.7 macrophages after treatment with 50% ethanolic extract *P. odoratum* Lour.cv. Pakphai leaves at various concentrations for 24 h. Data obtained from three independent experiments are expressed as mean ± SEM values. Accordingly, * *p* < 0.05 was considered significant when compared to control cells without LPS induction; # *p* < 0.05 was considered significant when compared to LPS-induced cells without the extraction treatment. Abbreviations: COX-2 = cyclooxygenase 2, LPS: lipopolysaccharide, SEM = standard error of the mean values.

**Figure 6 molecules-27-03657-f006:**
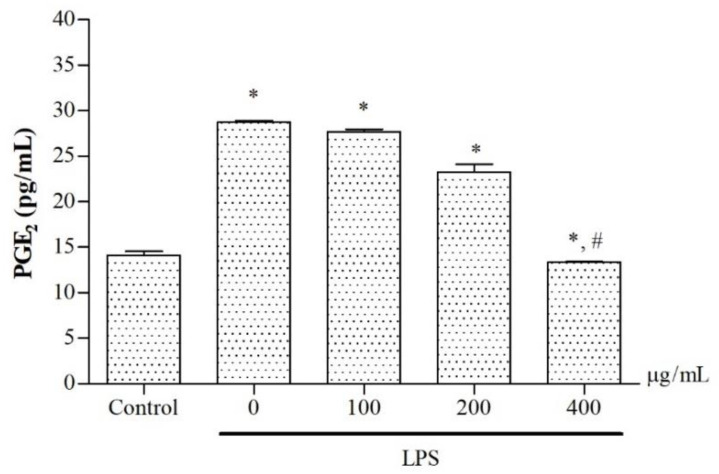
Levels of PGE_2_ production in LPS-induced RAW264.7 macrophages after being treated with 50% ethanolic extract of *P. odoratum* Lour. cv. Pakphai leaves at various concentrations for 24 h. Data obtained from three independent experiments are expressed as mean ± SEM values. Accordingly, * *p* < 0.05 was considered significant when compared to control cells without LPS induction; # *p* < 0.05 was considered significant when compared to LPS-induced cells without the extraction treatment. Abbreviations: COX-2 = cyclooxygenase 2, LPS: lipopolysaccharide, SEM = standard error of the mean values.

**Figure 7 molecules-27-03657-f007:**
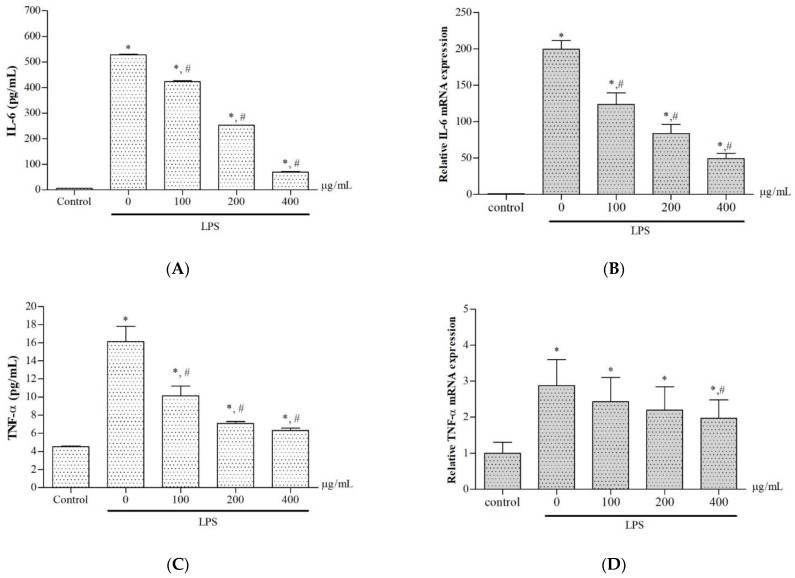
Levels of IL-6 production (**A**) and mRNA expression (**B**), as well as TNF-α production (**C**) and mRNA expression (**D**), in the medium obtained from the cultured RAW264.7 macrophage induced by LPS and treated with 50% ethanolic extract of *P. odoratum* Lour. cv. Pakphai leaves. Data obtained from three independent experiments are expressed as mean ± SEM values. Accordingly, * *p* < 0.05 was considered significant when compared to the control cells without LPS induction; # *p* < 0.05 was considered significant when compared to LPS-induced cells without the extraction treatment. Abbreviations: LPS = lipopolysaccharide, SEM = standard error of the mean values, TNF-α = tumor necrotic factor-alpha.

**Table 1 molecules-27-03657-t001:** UHPLC–ESI-QTOF-MS/MS identification of phenolic compounds in 50% ethanolic extract of *P. odoratum* Lour. cv. Pakphai leaves.

Peak No.	T_R_ (min)	Mode of Ionization	Estimated Mass (*m/z*)	Exact Mass [M] (g/mole)	Molecular Formula	Error (ppm)	Major MS/MS Fragments	Compounds	Chemical Structure
1	0.356	-	-	-	-	-	-	unidentified	-
2	0.677	-	-	-	-	-	-	unidentified	-
3	0.913	-	-	-	-	-	-	unidentified	-
4	2.112	-	-	-	-	-	-	unidentified	-
5	9.455	[M—H]^−^	478.07	478.36	C_21_H_18_O_13_	0.41	301.5, 162.1	Quercetin 3′-*O*-glucuronide	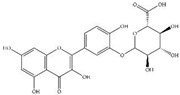
6	11.016	[M—H]^−^	462.08	462.40	C_21_H_18_O_12_	0.21	317.0, 147.0	3-Methylellagic acid 8-rhamnoside	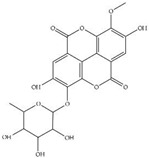
7	14.787	[M—H]^−^	302.04	302.23	C_15_H_10_O_7_	0.66	285.0, 193.0, 109.0	5,7,8,3′,4′-Pentahydroxyisoflavone or Q	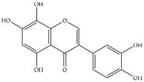
8	17.112	-	-	-	-	-	-	-	-
9	17.312	[M—H]^+^	291.21	290.26	C_15_H_14_O_6_	0.02	287.0, 269.9	EC	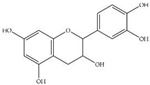
10	18.242	[M—H]^−^	305.24	306.27	C_15_H_14_O_7_	0.34	125.0, 137.0	GC	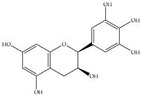
11	20.468	-	518.13	-	-	-	-	unidentified	-
12	21.531	[M—H]^−^	609.16	610.19	C_28_H_34_O_15_	0.65	304.2, 157.1	Hesperidin	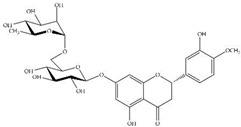
13	22.146	-	-	-	-	-	-	-	-
14	22.761	[M—H]^−^	121.09	121.18	C_8_H_11_N	3.30	109.0	2,6-Dimethylaniline	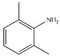

**Table 2 molecules-27-03657-t002:** HPLC-DAD analysis of the phenolic content in 50% ethanolic extracts of *P. odoratum* Lour. cv. Pakphai leaves. Data obtained from two separate experiments are expressed as mean ± standard errors of mean (SEM) values.

Peak No.	T_R_ (min)	Compounds	MW (g/mol)	Amount	
				(μg/mL Extract)	(μg/mg Dry Extract)
2	7.02	GA	170.12	3.85 ± 1.17	1.93 ± 0.04
3	12.27	C	290.26	1.70 ± 0.83	0.85 ± 0.04
4	14.00	EGCG	458.37	ND	ND
5	15.40	EC	290.26	ND	ND
6	17.28	ECG	442.37	10.85 ± 5.15	5.43 ± 0.01
7	21.27	CO	146.14	7.91 ± 0.67	3.95 ± 0.08
8	26.48	Q	302.24	13.47 ± 6.11	6.73 ± 0.01

Abbreviations: C = catechin, CO = coumarin, EC = epicatechin, ECG = epicatechin gallate, EGCG = epigallocatechin 3-gallate, GA = gallic acid, HPLC-DAD = high-performance liquid chromatography-diode array detector, MW = molecular weight, ND = not detectable, Q = quercetin, TPC = total phenolic content, T_R_ = retention time.

**Table 3 molecules-27-03657-t003:** Primer sequences for pro-inflammatory mediator expression in qRT-PCR.

Gene	Forward	Reverse
COX-2	TTCCTCTACATAAGCCAGTGA	TCCACATTACATGCTCCTATC
iNOS	AAGGTCTACGTTCAGGACATC	AGAAATAGTCTTCCACCTGCT
IL-6	GAGGATACCACTCCCAACAGACC	AAGTGCATCATCGTTGTTCATACA
TNFα	CATCTTCTCAAAATTCGAGTGACAA	TGGGAGTAGACAAGGTACAACCC
GAPDH	TGTGTCCGTCGTGGATCTGA	CCTGCTTCACCACCTTCTTGA

Abbreviations: A = adenine, C = cytosine, COX-2 = cyclooxygenase-2, G = guanine, GAPDH = glyceraldehyde-3-phosephate dehydrogenase, iNOS = inducible nitric oxide synthase, IL-6 = interleukin-6, T = thiamine, TNF-α = tumor necrosis factor-alpha.

## Data Availability

The data presented in this study are available on request from the corresponding author.

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
