# Peer review of "Quercetin-Rich Ethanolic Extract of Polygonum odoratum var Pakphai Leaves Decreased Gene Expression and Secretion of Pro-Inflammatory Mediators in Lipopolysaccharide-Induced Murine RAW264.7 Macrophages"

_molecules, 2022, doi:10.3390/molecules27123657_

Round 1

Reviewer 1 Report

I have gone through the manuscript entitled „ Quercetin-Rich Ethanolic Extract of Polygonum odoratum 2 var Pakphai Leaves Decreased Gene Expression and 3 Secretion of Pro-inflammatory Mediators in 4 Lipopolysaccharide-Induced Murine RAW264.7 5 Macrophages”. The article refers the important and current problem, therapeutic goal in the treatment of inflammatory disorders, and his medical implications cause that it contains in the thematic profile of the periodical Molecules. The work has an experimental character, carrying in relatively new cognitive elements from the sphere of basic sciences.  

RAW 264.7 cell line is a murine blood macrophages line and has been used to study the anti-inflammatory effects of pristimerin. It has also been used to test the internalization and intracellular survival of Listeria monocytogenes. Lipopolysaccharide  is the most abundant component within the cell wall of Gram-negative bacteria. It can stimulate the release of interleukin 8 (IL-8, CXCL8, CXC ligand 8) and other inflammatory cytokines in various cell types, leading to an acute inflammatory response towards pathogens.

 The Authors have convincingly proved that 50% ethanolic extract of P. odoratum Lour. cv. Pakphai leaves, non-toxic for murine blood macrophages, shows a relatively complex mixture containing peak quantities of phenolics, flavonoids and lignans ( high levels of quercetin, epicatechin gallate, coumarin, gallic acid and catechin). the leaves contain high amounts of bioactive ingredients that exhibit antioxidant, anti-hemolytic, potent anti-cancer, anti-aging and anti-bacterial properties.

The manuscript submitted to me for evaluation presents a study interesting and significant both for its molecular and clinical aspects. In general, title and design are appropriate, accompanied by the methodology correctly chosen, accurately described and applied. Only 44% of the cited literature comes from less than 5 years, and over 29% of the items are older than 10 years.

A holistic approach to treatment involves combining modern medicine with other fields of science, including dietetics, biochemistry, biomechanics or pharmacology. Interdisciplinary cooperation works well in everyday clinical practice, bringing excellent results. The research carried out for the purposes of this publication can help in the therapy of potential patients who will benefit from medical intervention related to inflammatory disorders.

Minor comments:

  • Typographical error: “p < 005” (line 196)
  • This kind of work usually ends with a conclusion about the need for further research (e.g. human cell lines), so it's good if they indicate exactly what research and for what reason.

Reviewer 2 Report

The study Quercetin-Rich Ethanolic Extract of Polygonum odoratum var Pakphai Leaves Decreased Gene Expression and Secretion of Pro-inflammatory Mediators in Lipopolysaccharide-Induced Murine RAW264.7 Macrophages has highlighted the determination that the ethanolic extract of Polygonum  odoratum Lour. cv. Pakphai leaves contained bioactive phenolics such as gallic acid, catechin and epicatechin 3-gallate; as well as flavonoids with a high content of quercetin and  coumarin.The findings indicate the anti-inflammatory activity of  this plant through the attenuation of a pro-inflammatory mediator response in macrophage cells.

There are Grammatical, spelling erros,  punctuation errors and also plagiarism is detected

Also writing issues are also detected, correct all these in the manuscript

Otherwise the planning of the experiment and execution is fine.

Major issue is characterizing the individual components in the ethanolic extract can be added.

Reviewer 3 Report

Corrections/suggestions to be attended as marked in the manuscript 

Self citations given are wrongly cited with respect to the references list. Eg Reference 15 and 20 are not authors' previous publications as quoted in the text. As such you have look at the whole reference list against the quoted references in the text. 

Round 2

Reviewer 2 Report

The modified version of the manuscript is ok